# POLYRETRO: FEW-SHOT POLYMER RETROSYNTHESIS VIA DOMAIN ADAPTATION

## ABSTRACT

Polymers appear everywhere in our daily lives – fabrics, plastics, rubbers, etc. – and we could hardly live without them. To make polymers, chemists develop processes that combine smaller building blocks (monomers) to form long chains or complex networks (polymers). These processes are called polymerizations and will usually take lots of human efforts to develop. Although machine learning models for small molecules have generated lots of promising results, the prediction problem for polymerization is new and suffers from the scarcity of polymerization datasets available in the field. Furthermore, the problem is made even more challenging by the large size of the polymers and the additional recursive constraints, which are not present in the small molecule problem. In this paper, we make an initial step towards this challenge and propose a learning-based search framework that can automatically identify a sequence of reactions that lead to the polymerization of a target polymer with minimal polymerization data involved. Our method transfers models trained on small molecule datasets for retrosynthesis to check the validity of polymerization reaction. Furthermore, our method also incorporates a template prior learned on a limited amount of polymer data into the framework to adapt the model from small molecule to the polymer domain. We demonstrate that our method is able to propose high-quality polymerization plans for a dataset of 52 real-world polymers, of which more than 50% successfully recovers the currently-in-used polymerization processes in the real world.

## 1 INTRODUCTION

Human beings are living in a world of chemical products, among which a category of chemicals, called polymers, is playing an essential role. Ranging from fabrics to plastics to rubbers, polymers are appearing in every corner of our daily lives. Polymers with different properties are desired when used in different circumstances, and chemists have been spending tremendous effort to design and synthesize new polymers in the pursuit of ones with better properties. To make polymers, chemists develop processes that combine small building blocks, which we call monomers, to form longer chains or complex networks. Such processes are called polymerization and will take a significant amount of human effort to develop.

Since the rise of deep learning (LeCun et al., 2015), applying these models to science problems like biology and chemistry ones have gradually gathered attentions. Specifically, the applications of AI methods in the retrosynthetic design of chemical compounds have become very popular recently (Segler et al., 2018; Coley et al.). While most work focuses on synthesizing drug-like small molecules, the study of polymer retrosynthesis is still at its infancy. The reasons are multifold, but one of the most important ones being the lack of available polymerization datasets, which poses difficulties for existing learning-based methods to learn meaningful pattern for polymerization reactions. Moreover, polymers usually have a chain or network structure with repeat units, which is very different from small molecules. This additional constraints also introduces difficulties in the formulation and modeling of polymer design/retrosynthesis.

In this paper, we focus specifically on the polymer retrosynthesis problem. While there has been a series of work focusing on small molecule retrosynthesis (Corey & Wipke, 1969; Gasteiger et al., 1992; Coley et al., 2017; Liu et al., 2017; Segler & Waller, 2017; Segler et al., 2018; Coley et al.;

Karpov et al., 2019; Baylon et al.; Schreck et al.; Dai et al., 2019; Chen et al., 2020), the problem of polymerization is very different and challenging in the machine learning sense that

1. To predict synthesis routes for polymer repeat units, additional structural constraints such as recursive constraint should be imposed to guarantee a potentially valid polymerization procedure. Such constraints do not exist in existing formulation for molecule retrosynthesis, thus most methods could not be directly applied.
2. Polymerization data for training is very limited. Compared with retrosynthesis models built for small molecules where accessible training data is at least tens of thousands, the size of polymerization data is tiny, and in our case it is even less than 100. This size is meaningless for most existing models to learn any synthesis patterns.

In this paper, we formulate the problem of polymer retrosynthesis as a constrained optimization and present `PolyRetro`, a novel learning-based search framework to tackle the problem of polymer retrosynthesis. With beam search and rejection sampling, `PolyRetro` is able to propose monomer candidates with high polymerization probability while satisfying monomer synthesizability constraints. Our method is based on reaction templates collected from small molecule reactions, which capture the local structural properties for small molecule reactions. We leverage an one-step retrosynthesis model trained on a small molecule reaction dataset and adapt it to the polymer domain by incorporating a template prior learned on tiny-sized polymerization data. To verify whether the proposed monomers are synthesizable, we employ Retro* (Chen et al., 2020), a multi-step retrosynthesis model to predict their synthesis routes. We demonstrate `PolyRetro` through experiments that it is able to predict monomers accurately given target polymer repeat units. To our knowledge, we are the first to formulate, model and tackle the polymer retrosynthesis machine learning problem.

The approach we developed is general in the sense that it can also be applied to other machine learning problem such as theorem proving and program synthesis, where the results we want to obtain involves recursion. For instance, the analogue problem in theorem proving is deriving proof for a theorem which contains recursive relations; the analogue problem in program synthesis is generating programs containing loops and recursive calls. We choose to focus our application in polymer synthesis because its importance and high societal impact. See Appendix E for more concrete discussions on the generality of `PolyRetro`.

Our contributions are summarized below:

- We formulate the problem of polymer retrosynthesis as a constrained optimization problem. To our knowledge, this is the first machine learning formulation that takes constraints in polymer retrosynthesis into consideration.

- We propose PolyRetro, a learning-based search framework that tackles the problem of polymer retrosynthesis. To our knowledge, this is also the first learning-based method in this problem setting.

- PolyRetro is able to recover $53\%$ of ground truth monomers for a real-world polymer dataset using limited training data, significantly outperforming all existing algorithms.

## 2 RELATED WORKS

Computer-aided retrosynthetic planning for chemical molecules was first formalized by E. J. Corey (Corey & Wipke, 1969) and have been deployed over the past years. The task of retrosynthetic design is to identify *a series of reactions* that leads to the synthesis of target molecule. This is one of the most fundamental problems in organic chemistry.

Recently, many machine learning methods has been proposed to the easier but also important subproblem, where one is given target molecule and the task is to predict the direct predicates (Coley et al.). Methods to tackle such 'one-step version' of retrosynthesis could be roughly divided into two categories, template-based and template-free ones. A template of chemical reaction is essentially how bonds and atom change during the reaction, and could be applied reversely to get reactants from products. Thus there have been a series of methods trying to predict the reaction templates given product molecules to get the corresponding reactants (Coley et al., 2017; Segler & Waller, 2017; Baylon et al.; Dai et al., 2019). While powerful, these methods are not applicable in the case where training data comes without templates. To resolve this, there have been attempts to use

sequence-to-ssequence model to directly predict SMILES [1] representation of reactants (Liu et al., 2017; Karpov et al., 2019). Such methods could be straight-forwardly applied to any reaction data, but may need a large number of reactions for training to find meaningful reaction patterns without help of reaction templates.

On the other hand, there has been works trying to directly solve multi-step retrosynthesis prediction (Segler et al., 2018; Schreck et al.; Kishimoto et al., 2019; Chen et al., 2020). This multi-step procedure is usually decomposed into two modules, a one-step retrosynthesis module which could propose possible direct predicates given product molecules, and a planning algorithm to search for best synthetsis route with recursive application of one-step module.

While there has been such rich literature in the retrosynthesis domain, a paucity of work has tackle the specific problem of polymer retrosynthesis. We focus on polymer retrosynthesis and bring knowledge from general retrosynthesis literature to help tackle the problem.

## 3 BACKGROUND

In this section, we focus on providing the background knowledge about molecule retrosynthesis as well as defining notations. This serves as the building block in our polymer retrosynthesis modeling.

Given a molecule $m \in \mathcal{M}$ where $\mathcal{M}$ indicates the space of molecules, the molecule retrosynthesis problem focuses on finding a set of reactants $\mathcal{S} \subset \mathcal{M}$ that can be used to synthesize $m$. Before introducing the approaches for retrosynthesis, we first cover the background on reaction templates.

### 3.1 REACTION TEMPLATE

A reaction template $T := o^T \to r_1^T + r_2^T + \ldots + r_{|T|}^T$ is a graph rewriting rule [2] that rewrites subgraph pattern $o^T$ that is matched with target molecule $m$, into $r_i^T$ that appears in $i$-th reactant $s_i \in \mathcal{S}$. The set of templates $\mathcal{T}$ can be extract from existing chemical reactions in the literature. Although applying templates involves with expensive subgraph matching between $o^T$ and $m$, where itself is an NP-hard problem, such approach provides a tractable way of finding candidate set $S$ with chemical rules.

### 3.2 LEARNING-BASED MOLECULE RETROSYNTHESIS

The molecule retrosynthesis problem has raised increasing interest in the machine learning (ML) community, due to its importance in chemistry and the difficulty in structured prediction setting. We here mainly focus on the ML approaches for such problem, as some of these provide probabilistic interpretations that will be needed in our optimization framework. Depends on the number of reaction steps needed to synthesize $m$, such problem can be categorized into one-step and multi-step retrosynthesis.

**One-step molecule retrosynthesis:** One-step setting requires that $R := \mathcal{S} \to m$ can be realized in one chemical reaction. It focuses on modeling $p(\mathcal{S}|m)$ with or without reaction templates. As the template based one guarantees the satisfaction of human defined rules, we use the model proposed in NeuralSym (Segler & Waller, 2017) for one-step prediction. Specifically in this model:

$$p(\mathcal{S}|m) \propto \sum_{T \in \mathcal{T}} p(T|m) \mathbb{I} \left[ \text{SubgMatch}(o^T, m) \right] \tag{1}$$

where SubgMatch($\cdot$) operator checks the subgraph matching between $o^T$ and $m$.

**Multi-step molecule retrosynthesis** The multi-step extension allows using multiple reactions $\mathcal{R}_m = \{R_i^m\}_{i=1}^{|\mathcal{R}_m|}$ to synthesize $m$, with the restriction that the reactants set $\mathcal{S} \subset \mathcal{I} \subset \mathcal{M}$ where $\mathcal{I}$ is the set of starting molecules. This is essentially a planning problem that search through the reaction space using one-step models as expansion proposals. In our paper, we use the Retro*(Chen et al., 2020) which is the state-of-the-art approach that provably optimizes the synthesizability of $\mathcal{R}_m$.

---

[1] https://www.daylight.com/dayhtml/doc/theory/theory.smiles.html.

[2] https://www.daylight.com/dayhtml/doc/theory/theory.smarts.html

Figure 1: Condensation polymerization for synthesizing PET. We use shaded color regions to highlight the monomers (light blue), polymer (light pink), repeat unit (dark pink), end-groups (purple), and by-product (light yellow). Unit polymer is a polymer with unit length ($n = 1$).

## 4 MODELING POLYMER RETROSYNTHESIS

A polymer is a large molecule composed of many repeat units. Directly using molecule retrosynthesis techniques for polymers is not feasible, as **(i)** the (potentially infinite) large molecule is not feasible for existing molecule retrosynthesis approaches; **(ii)** the polymer retrosynthesis has nontrivial recursive and stability contraints, which cannot be easily addressed in existing approaches.

In the following content, we first state these constraints in Section 4.1, then we formally present our modeling strategy for such problem in Section 4.2.

### 4.1 CONSTRAINTS FOR POLYMERS AND POLYMERIZATIONS

In chain structured polymer, each repeat unit in the polymer has two open bonds which link with neighboring units in a chain-like structure. At the end of the chain are the end-groups. They are functional groups closely related to the polymerization process, where the polymer is synthesized from the monomers.

In this work we focus on **condensation polymerization**, in which large molecules join together and lose small molecules such as $H_2O$/HCl (Figure 1). This type of polymerization needs to satisfy the recursive constraint (unit polymerization) placed on the unit polymer.

**Definition 1** *(Recursive constraint) Given the repeat unit $r$ with structure -A-, where we use '-' to represent an open bond, which connects to the neighboring repeat unit in a long chain, the unit polymer $u$ should be b-A-c, where b- and -c are end-groups, and the probability of the following induced unit polymerization reaction should be positive:*

$$\text{UnitPolymerization}(r, u) : \text{b-A-c} + \text{b-A-c} \rightarrow \text{b-A-A-c} + \text{b-c} \tag{2}$$

In this reaction, two unit polymers join together and lose b-c as the byproduct. In practice, we hope that the end-groups b- and -c tend to react with each other with high probability, which makes the polymerization reaction continue to happen. Therefore the unit polymer itself is not stable. However the monomers, which are the precursors of the unit polymer, should satisfy the stability constraint.

**Definition 2** *(Stability constraint) Given the unit polymer $u$ with structure b-A-c, the monomers $m$ should be {b-B-b, c-C-c}, where b-B-b and c-C-c are symmetric, and $u$ can be synthesized from $m$ with the following reaction:*

$$\text{UnitPolymerSynthesis}(u, m) : \text{b-B-b} + \text{c-C-c} \rightarrow \text{b-A-c} + \text{b-c} \tag{3}$$

Both of the above constraints come from chemical insights which guide polymer synthesis.

### 4.2 POLYMER RETROSYNTHESIS MODELING

Section 4.1 has defined the structural constraints for polymer retrosynthesis, which are symbolic with rule definitions. In real application, polymerization also requires the efficiency to make the entire pipeline cost efficient. With both the rule constraints and efficiency requirement, we formulate the polymer retrosynthesis as a constrained optimization problem. We first define molecule synthesizability below:

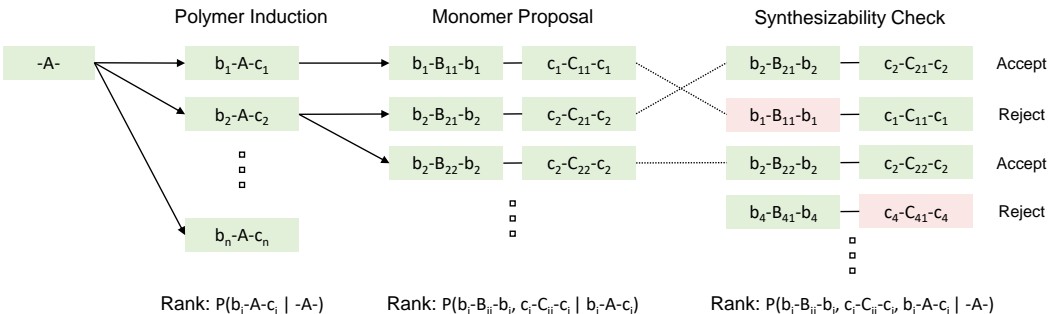

Figure 2: PolyRetro beam search framework. Given the repeat unit $r$, we first generate unit polymer candidates $u$ by polymer induction. We then rank the candidates using $p(u|r)$ and sample monomers $(m_1, m_2)$ from the top-$n$ unit polymers. Next we re-rank the unit polymer and monomer pair $(u, m_1, m_2)$ using the joint $p(u, m_1, m_2|r) = p(u|r) \cdot p(m_1, m_2|u)$ and perform the synthesizability check on $(m_1, m_2)$ using Retro*, a multi-step retrosynthesis planner. Finally the top-$k$ results which pass the check are returned.

**Definition 3** *(Molecule synthesizability) A molecule $m$ is $h$-synthesizable from a set of starting molecules $\mathcal{I}$ if and only if there exists a series of reactions $\mathcal{R}$ whose initial reactants are in $\mathcal{I}$, and the joint probability of reactions in $\mathcal{R}$ satisfy a given lower bound $h$, i.e. $p(\mathcal{R}) > h$.*

Practically we rely on the molecule retrosynthesis algorithm presented in 3.2 to obtain both $\mathcal{R}$ and $p(\mathcal{R})$. Using the key concepts defined above, we state our optimization formulation below:

**Definition 4** *(Polymer Retrosynthetic Optimization) Given a target polymer represented as a repeat unit $r$, we want to identify a unit polymer $u$, and monomers $m_1$, $m_2$ which maximize the polymerization probability with the recursive, stability, and molecule synthesizability constraints.*

$$
\begin{aligned}
\max_{u, m_1, m_2} \quad & p(u, m_1, m_2|r) = p(u|r) \cdot p(m_1, m_2|u) \\
\text{s. t.} \quad & p(\mathcal{R}_{m1,m2}) > h. \text{ where } h > 0
\end{aligned}
\tag{4}
$$

The model $p(u, m_1, m_2|r)$ is decomposed into $p(u|r)$ that incorporates the recursive constraint, and $p(m_1, m_2|u)$ that incorporates the stability constraint. The formulation can be interpreted using a mathematical induction analogy. Synthesizing the target polymer can be understood as proving the feasibility of the polymer. To "prove" the target polymer, we need to first work on the base case, *i.e.*, find a monomer which is synthesizable, and then prove the induction step, *i.e.* maximize the polymerization probability.

Although $p(m_1, m_2|u)$ can be characterized using one-step molecule retrosynthesis model, directly optimizing the above problem is nontrivial, as **(i)** the recursive constraint model $p(u|r)$ which predicts a unit polymer from a repeat unit is not available and there is not enough data in the literature to estimate such model; and **(ii)** the constraint of $p(\mathcal{R}_{m1,m2}) > h$ goes through a planning algorithm that is not possible to characterize the gradient, convexity, *etc.*. In the next section, we will present `PolyRetro` to effectively solve the constrained optimization by overcoming these difficulties.

## 5 POLYRETRO FOR POLYMER RETROSYNTHESIS

To tackle the challenging constrained optimization problem defined in Section 4.2, we propose `PolyRetro` (Figure 2), a learning-based search framework with minimal polymerization data involved. We first introduce the overall procedures of the algorithm, and then cover the details.

Our main idea is based on the rejection sampling framework for solving Eq (4), which treat the molecule synthesizability constraint as a black-box rejection criteria. Although asymptotically we can sample target solution $(u, m_1, m_2)$ from Eq (4), a good proposal algorithm is needed to keep the rejection rate low and mix fast. Since the joint probability of $(u, m_1, m_2)$ decomposes into two

terms, we approximate the max operator with a beam search[3] in two steps:

$$\texttt{top-k}_{u,m_1,m_2} p(u, m_1, m_2 | r) \simeq \texttt{top-k}_{u,m_1,m_2} p(m_1, m_2 | u) \mathbb{I}\left[u \in \texttt{top-n}\left\{p(u|r)\right\}\right] \quad (5)$$

We present the modeling and inference of these two steps in the next two sections. In Section 5.1 we show how to generate top-$n$ candidate unit polymers that satisfy with recursive constraint using domain adaptation with a novel *polymer induction* technique. Then in Section 5.2, we show the generative procedure of monomers with stability constraints. We conclude the approach with the molecule synthesizability check in Section 5.3.

## 5.1 Unit polymer proposal with domain adapted polymer induction

In the first step of the beam search, we want to quickly generate all the feasible candidate unit polymers from the repeat unit $r$. Proposing candidates that satisfy recursive constraint is hard. This involves predicting the end-groups from the repeat units. Also note that there will not be enough data to directly learn the model $p(u|r)$.

We observe that, it is relatively easy to obtain the one-step retrosynthesis model (one-step model for short) $p(\mathcal{S}|m)$ in Eq (1) for a *molecule* $m$ and there are enough small molecule chemical reactions to train such one-step model. Based on this, we propose to perform *domain adaptation* using *polymer induction* technique (Figure 3), as explained below:

**Polymer induction**: the basic idea is to leverage one-step molecule retrosynthesis model to help predict the end groups of a unit polymer. As the model $p(\mathcal{S}|m)$ only accepts molecules rather than repeat units, we circumvent this issue with the following procedure:

1. Link two repeat units head-to-tail to form a double repeat unit, and add hydrogen as end-groups, i.e. H-A-A-H;
2. Loop through all the reaction templates:
   (a) Apply reaction template to the double repeat unit;
   (b) If the result is in the form of H-A-A-H → H-A-c + b-A-H, add b-A-c to the candidate list.

The above procedure is based on the induction principle: if the base case solves (*i.e.*, we find the unit polymer with form b-A-c), then we can synthesize the double units b-A∼A-c where the bond between two As (denoted as ∼ symbol) belongs to the reaction center. Such induction thus provides the guidance for the template based retrosynthesis: a template $T = o^T \rightarrow r_1^T + r_2^T$ should have the subgraph $o^T$ matched at the location that covers the bond denoted as ∼. Using template based graph rewriting with $r_1^T$ and $r_2^T$, we can obtain the end groups $b$ and $c$.

Figure 3: Polymer induction for generating unit polymer candidates. For each of the reaction template, we apply it to the double repeat unit, if it can break the bond connecting two repeat units into two end-groups. These two end-groups are then being put back to the repeat unit to reconstruct unit polymer candidates.

**Modeling $p(u|r)$ with domain adaptation**: the polymer induction step gives us a list of unit polymer and corresponding template candidates $\{(T_i, b_i\text{-A-}c_i)\}$ that satisfy the recursive constraint, which is actually the support of $p(u|r)$ (*i.e.*, samples with non-zero probability). Given a unit polymer and template pair $(u = \text{b-A-c}, T)$, we formulate the corresponding joint probability as the optimal solution of following optimization:

$$\min_{p(u=\text{b-A-c},T|r)} \lambda \underbrace{D_{KL}(p(\mathcal{S} = \{\mathbf{u}, \mathbf{u}\} | m = \text{b-A-A-c}) || p(u, T|r))}_{\text{matching one-step model}} + (1 - \lambda) \underbrace{D_{KL}(\hat{p}(T) || p(u, T|r))}_{\text{matching target domain prior}}$$

$$(6)$$

where $\hat{p}(T)$ is an empirical estimation that of template prior from the limited polymer dataset, and $\lambda \in (0, 1)$ is a factor tha balances the two Kullback–Leibler divergences. The optimal solution is

---

[3]Beam search has similar performance as other sampling schemes when $k$ is large, as we show in Appendix D.

$p(u = \text{b-A-c}, T|r) = \lambda p(\{u, u\} |\text{b-A-A-c}) + (1 - \lambda)\hat{p}(T)$. This objective performs the domain adaptation from molecule synthesis into polymer synthesis with the following benefits:

- With the polymer induction technique, we adapt one-step model for molecules to polymer domain.
- By interpolating between one-step model and the prior learned on target domain, we can achieve a balance between enormous foreign domain knowledge and limited target-domain priors.

There could be multiple design choices for $\hat{p}(T)$. Due to the limited data in target (polymer) domain, we use kernel density estimation with atom- and bond-counting features for simplicity.

## 5.2 MONOMER PROPOSAL UNDER STABILITY CONSTRAINT

Using domain adapted polymer induction in the above section, we can obtain top-$n$ unit polymers $\{b_i\text{-A-}c_i\}_{i=1}^n$ and their scores. The second step of the beam search seeks for monomers given the unit polymers. As both of them are proper molecules, the one-step model can be directly used here:

$$p(m_1, m_2|u) = p(\mathcal{S} = \{m1, m2\} |m = u)\mathbb{I}\left[\text{Both } m_1 \text{ and } m_2 \text{ are symmetric}\right] \quad (7)$$

Adding together with the scores of each unit polymer $u$, we can approximately get a list of feasible unit polymer and monomers with the highest joint probability.

## 5.3 SYNTHESIZABILITY AS FILTERING CRITERIA

Once we have the top-$n$ list of monomers $\{m_1^i, m_2^i\}_{i=1}^n$ obtained in previous two beam-search steps, we can perform rejection step using any off the shelf multi-step retrosynthesis planner to check the synthesizability $p(\mathcal{R}_{m_1,m_2})$ for each $(m_1, m_2) \in \{m_1^i, m_2^i\}_{i=1}^n$, and return the top-$k$ of them which satisfies the recursive, stability and synthesizablity constraints. This finishes `PolyRetro` algorithm.

# 6 EXPERIMENTS

**Experiment Settings.** To evaluate the performance of `PolyRetro`, we collect a dataset of 52 condensation polymers[4], and their corresponding synthesis recipes. For each polymer, the task is to predict the ground truth monomers and unit polymer given the repeat unit. We split the data into 5 folds and repeat the same experimental setup 5 times to compute the mean and standard deviation of the performance metric. Because we are targeting a few-shot learning setting and considering we do not have enough data for testing, in each experiment, we only use one fold of data for training and hold out the other four folds for evaluation. We use top-$k$ recovery rate, i.e. whether the target is in the top-$k$ prediction, on both unit polymers and monomers to measure the performance.

**`PolyRetro` Implementation.** The one-step retrosynthesis model used in `PolyRetro` is trained on the publicly available reaction dataset extracted from United States Patent Office (USPTO). We use a multilayer perceptron model with one hidden layer of size 512, which takes a 2048 dimensional molecular fingerprint (Rogers & Hahn, 2010) as input, and predicts a probability distribution over 230k reactions templates extracted from the same dataset. Reactants can be obtained by applying the reaction templates to the product using RDKit (rdk). In the monomer retrosynthesis step, Retro* (Chen et al., 2020) employs the same model for synthesis route search given a list of commercially available building blocks from *eMolecules*[5], which consists of $231M$ commercially available molecules.

We implement the template prior in `PolyRetro` using a nonparametric Gaussian kernel density estimator with bandwidth 1. Its performance is not sensitive to the bandwidth parameter. The features are the number of atoms and edges on both side of the template. We set $\lambda$ to 0.999 in Eq (6).

**Baselines/Ablation Studies.** We compare `PolyRetro` with a Transformer (Vaswani et al., 2017) model which directly predicts the target sequence (unit polymers/monomers) from the repeat unit. We also conduct an ablation study by replacing the unit polymer proposal step in `PolyRetro` with (a) sampling randomly from the support of $p(u|r)$ (Random Proposal), and (b) using only the one-step retrosynthesis model trained on USPTO as $p(u|r)$ (`PolyRetro`-USPTO). Since learning a sequence-to-sequence model from only a few data points is nearly impossible, we give all Transformer-based models a large advantage by allowing the use of test data for validation for early stopping in training.

---

[4]It's hard to collect data in this domain. We conduct human evaluations on an additional 100 polymers which have no ground truth recipes. More details are discussed in Appendix D.

[5]http://downloads.emolecules.com/free/2019-11-01/

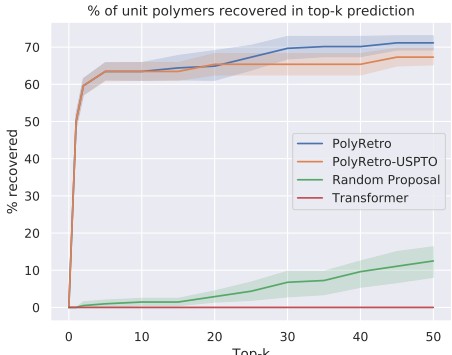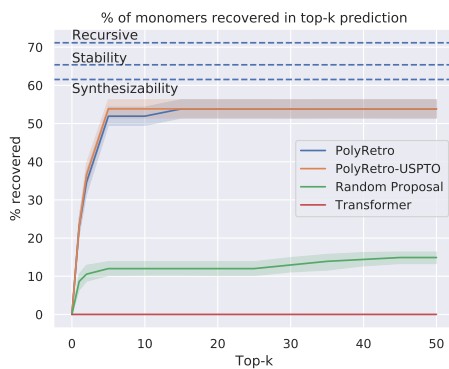

Figure 4: Percentages of targets recovered by algorithms in top-$k$ prediction. Left: unit polymers prediction using $p(u|r)$; Right: monomer prediction using $p(u|r) \cdot p(m_1, m_2|u)$. We also show `PolyRetro`'s performance upper bounds after addressing the labeled constraints on the right.

**Results.** The top-$k$ recovery rate result on unit polymers and monomers is summarized in Figure 4. `PolyRetro` achieves over $50\%$ top-1, $71\%$ top-30 recovery rate in unit polymer prediction, and over $50\%$ top-5 recovery rate in monomer prediction. The performance is significantly better than the sequence-to-sequence baseline, which is not able to learn anything, achieving $0\%$ top-50 recovery rate in both results, not to mention that the baseline model can see the test set during training. This is within our expectation due to the lack of training data. With more data we should see an increase in the number. However, end-to-end approaches still suffer from the inherent limitation of failing to address the structural constraints. When the baselines are given even more advantages by knowing structural priors in advance, they still cannot perform as well as `PolyRetro`, as shown in Appendix D.

**Domain Adaptation.** Knowledge transfer from small molecule reactions to unit polymerization is clearly supported by the result, as `PolyRetro` and `PolyRetro`-USPTO outperform the random proposal baseline by a large margin. With such strong performance of `PolyRetro` recovering the ground truth monomers, we have reason to believe the other monomers generated by our algorithm could also be of realistic significance. Domain adaptation by incorporating template prior is also helpful for improving the performance. With the template prior learned on one fold of data, `PolyRetro` is able to have a $3 - 5\%$ gain in performance compared with `PolyRetro`-USPTO when predicting unit polymers. The main reason that `PolyRetro` is not improving the monomer prediction further is because of the limitation on unseen templates for one-step retrosynthesis, which we discuss below.

**Limitations.** We display the performance upper bound of `PolyRetro` in the monomer prediction result in Figure 4. From top to bottom, the three horizontal lines correspond to the upper bounds after addressing the recursive constraint, the recursive and stability constraints, and all three constraints. These upper bounds are resulted from the following limitations of the algorithm.

- **Constraint Formulation.** Our constraint formulations in Definition 1 and 2 describe the most common pattern for polymerization but unable to cover all special cases. The algorithm can be improved by incorporating more chemical knowledge.
- **Unseen Reaction Templates.** Since the templates for one-step retrosynthesis model are extracted from the small molecule reactions, there could be new reaction templates which only exist in polymerization. The limitation can be alleviated by extracting templates and training the one-step retrosynthesis model directly using polymerization data.
- **Monomer Retrosynthesis.** For rare cases, Retro* cannot find a synthesis path for the ground truth monomer in limited time. This can be resolved by allowing more time for Retro* search and using a larger building block molecule set.

## 7 CONCLUSION

Polymer synthesis is an important yet challenging task with high societal impact. In this work, we propose a novel constrained optimization formulation for the problem, together with a learning-based search framework to tackle the problem. Our framework is able to handle the structural constraints that appeared in the polymerization process, and to deal with small data regimes by adapting the model trained on small molecule reaction dataset to polymer domain. Our method significantly outperforms the baselines in terms of the ground truth monomers recovered rate in top-$k$ prediction.

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

# A  POLYMERIZATION EXAMPLES

More examples of condensation polymerization are presented below. The recursive and stability constraints are strictly imposed here.

Figure 5: Condensation polymerization for synthesizing nylon-6,6 from hexamethylenediamine and adipic acid.

Figure 6: Condensation polymerization for synthesizing Kevlar from p-Phenylenediamine and terephthaloyl chloride.

# B  ALGORITHM SUMMARY

To attack the constrained optimization problem defined in Eq (4), `PolyRetro` (Algorithm 1) adopts a rejection sampling approach using a two-step beam search as the proposal distribution. The first step (line 1-2) of beam search proposes valid unit polymer candidates given the target repeat unit using domain adaptation with polymer induction. Then in the second step (line 3-5), monomer candidates are proposed given the unit polymers. Finally rejection sampling (line 6-9) is performed to filter out infeasible monomers.

---

**Algorithm 1:** `PolyRetro`$(r)$

1  Compute $p(u|r)$ via *domain adaptation* using *polymer induction* (Eq (6))
2  $\{u_i\}_{i=1}^n = \texttt{top-n}_u\, p(u|r)$
3  **for** $i \leftarrow 1$ **to** $n$ **do**
4      Compute $p(m_1, m_2|u_i)$ via one-step model with constraints (Eq (7))
5      $\{(m_1^{(i,j)}, m_2^{(i,j)})\}_{j=1}^t = \texttt{top-t}_{m_1,m_2}\, p(m_1, m_2|u_i)$
6  **for** $i \leftarrow 1$ **to** $n$ **do**
7      **for** $j \leftarrow 1$ **to** $t$ **do**
8          $\mathcal{R}_{m1,m2} = \texttt{Retro}^*(m_1^{(i,j)}, m_2^{(i,j)})$
9          $accept^{(i,j)} = p(\mathcal{R}_{m1,m2}) > h$
10 $\{(u^{(i)}, m_1^{(i)}, m_2^{(i)}, \mathcal{R}_{m1,m2}^{(i)})\}_{i=1}^k = \texttt{top-k}_{u,m_1,m_2}\, p(u|r) \cdot p(m_1, m_2|u) \cdot \mathbb{I}[accept^{(u,m1,m2)}]$
11 **return** $\{(u^{(i)}, m_1^{(i)}, m_2^{(i)}, \mathcal{R}_{m1,m2}^{(i)})\}_{i=1}^k$

---

## C  TEMPLATE PRIOR ESTIMATION

To estimate the template prior $\hat{p}(T)$ in Eq (6), we use kernel density estimation with Gaussian kernel,

$$p_{kde}(\boldsymbol{x}) = \frac{1}{n(2\pi)^{d/2}h^d} \sum_{i=1}^{n} \exp(-\frac{1}{2}(\frac{\boldsymbol{x} - \boldsymbol{x}^{(i)}}{h})^{\top}(\frac{\boldsymbol{x} - \boldsymbol{x}^{(i)}}{h})), \quad (8)$$

where $d = 6$, $\boldsymbol{x}^{(i)} = (na(o^{T_i}), nb(o^{T_i}), na(r_1^{T_i}), nb(r_1^{T_i}), na(r_2^{T_i}), nb(r_2^{T_i})) \in \mathbb{R}^d$ is the feature vector for the $i$-th template $T_i := o^{T_i} \rightarrow r_1^{T_i} + r_2^{T_i}$. Here we use $na(s)$ and $nb(s)$ to denote the number of atoms and bonds in subgraph $s$.

## D  MORE EXPERIMENTS

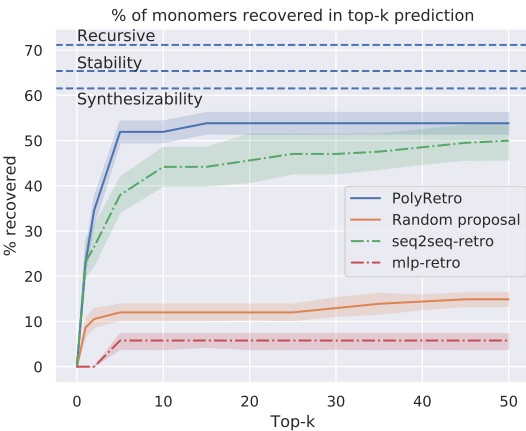

Figure 7: Top-$k$ monomer recovery with added baselines. The dash-dotted lines represent baselines with additional advantage.

**Additional Baselines.** We introduce two baselines, **seq2seq-retro** and **mlp-retro**, to monomer recovery experiments. Both are based on existing one-step retrosynthesis methods. Since it is infeasible to directly apply one-step models to the repeat unit $r$, which is an incomplete molecule, we **give additional advantage to both baselines** by completing $r$ using one of the most common end-groups pairs, and perform one-step retrosynthesis on the completed molecule H-$r$-H to obtain the monomer candidates. Details of two baselines (originally one-step models) are: **1)** seq2seq-retro: seq2seq model pretrained on USPTO, fune-tuned on the training split; **2)** mlp-retro: mlp model trained on USPTO. It is template-based and cannot be fine-tuned.

Although given additional advantage, both baselines are out-performed by `PolyRetro` (Figure 7). The mlp-retro baseline is not performing well, as it is based on reaction templates which cannot modify the fixed end-groups. The seq2seq baseline is doing well for a large $k$, but unlike `PolyRetro` and mlp-retro, it fails to provide chemical explanations.

|  | beam search | softmax sampling | nucleus sampling |
|---|---|---|---|
| $k = 2$ | $0.346 \pm 0.035$ | $0.284 \pm 0.051$ | $0.038 \pm 0.011$ |
| $k = 10$ | $0.519 \pm 0.028$ | $0.538 \pm 0.028$ | $0.519 \pm 0.023$ |

Table 1: Percentages of monomers recovered under different sampling schemes.

**Ablation Study on Sampling Schemes.** We explore other sampling schemes including softmax sampling and nucleus sampling. Both strategies lead to similar performance as the beam search we used in `PolyRetro` when $k$ is large enough.

**Evaluation on Additional Polymers.** We collect an additional 100 polymers for evaluating monomer prediction, and use the original 52 polymers for training. As we do not have the synthesis recipes for

these polymers, the evaluation can be seen as the process of proposing monomers for synthesizing a newly found polymer. We conduct a double-blind A/B testing similar to (Segler et al., 2018) by asking experienced chemists to compare the top-1 monomers proposed by different algorithms. We found that seq2seq-retro does not apply in this scenario: for all 100 polymers whose repeat units are much larger than the ones in the training set, seq2seq-retro predicts the same monomer as output. This is probably due to seq2seq models do not generalize well to longer and unseen sequences. The A/B testing result between `PolyRetro` and mlp-retro is: **1)** `PolyRetro` gives the more sensible solution for **82/100** polymers and **2)** all solutions for the rest **18/100** polymers have apparent flaws. Additionally, the chemists found that in most of the failed cases, the monomers do not have more than one functional groups which allow them to react recursively and form a chain.

## E  GENERALITY OF POLYRETRO

`PolyRetro` can be adapted to learn to solve combinatorial search problems where the search targets have additional recursive constraints. As an example, we will show how to apply it to searching symbolic programs for integer addition tasks (Kaiser & Sutskever, 2015; Trask et al., 2018; Cai et al., 2017). Integer addition of arbitrary length is an elementary skill to human. However, it is proven to be very difficult for standard neural networks, because it needs simultaneously learning of single digit operations and the concept of recursion. This can be formulated using `PolyRetro` if we view the search policy for a program for single digit operation as the monomer retrosynthesis model, and the recursion as the polymerization prediction step. In the few-shot learning case, domain adaptation would be useful to transfer knowledge across domains, such as integer addition in other bases, to the current one. Both the recursion and few-shot learning challenges can be handled by `PolyRetro`. Other examples include:

- learning a sorting program (Cai et al., 2017; Li et al., 2020), where we need to design a sorting algorithm with recursive structures that solve the sorting of sub-sequences, and adapt to domains with different "comparison function" needed for sorting;

- learning a shift-reduce algorithm for parsing (Dyer et al., 2016), where we need to search over shift/reduce polices in the recursive parsing algorithm, and adapt to domains with different grammars (e.g., different domain specific languages).

