# OpenReview forum: "PolyRetro: Few-shot Polymer Retrosynthesis via Domain Adaptation"
_ICLR.cc/2021/Conference — Reject_

### Official Review · AnonReviewer3 · 2020-10-25

**Rating:** 5
**Confidence:** 3

**Review:**

Summary
- This paper aims at solving an important yet challenging task, the polymer retrosynthesis problem.
- This paper formulates the condensation polymerization as a constrained optimization with prior structural knowledge.
- This paper proposes a framework to solve the problem and demonstrates its effectiveness in a few-shot benchmark.

---

Strengths
- Although I'm not an expert in chemistry, I already know that retrosynthesis is a very important problem in academia and industry. Thus, polymer retrosynthesis also seems to be practically important.
- This paper well explains how to apply a single-step retrosynthesis model (trained in a reaction benchmark) into the polymer retrosynthesis problem. All sub-steps (except domain adaptation) are reasonable and easy to understand.

---

Concern (or Weakness) \#1: Domain adaptation
- The domain adaptation is emphasized throughout this paper. However, I think this adaptation is hard to provide a meaningful gain in general, (a) fundamentally in the few-shot setting and (b) practically as shown in Figure 4 (right). Details of (a) and (b) are described below.
  - (a) In the few-shot setting, the prior empirical distribution $\hat{p}(T)$ is often meaningless. This is because the reaction templates are very diverse, e.g., 40k reactions in USPTO-50k has 10k different templates, so the distribution is often very sparse, especially in the few-shot setting. In this case, optimizing Equation (6) cannot improve $p(u|r)$. In my opinion, the authors' choice, using atom- and bond-counting features, might alleviate this issue, however, I'm not sure what is the (chemical) meaning of this simplified distribution, and it can generalize to other polymer benchmarks.
  - (b) As illustrated in Figure 4 (right), the adaptation does not provide any gain in the monomer prediction even though it depends on $p(u|r)$ as described in the caption of Figure 4. It means that the stability constraint will filter out some wrong predictions in $p(u|r)$ from PolyRetro-USPTO. It is then hard to say that the domain adaptation is helpful for polymer retrosynthetic optimization (Definition 4).
- Since tempalte-based models are used and $u$ is determined when $T$ is given, the PolyRetro model with domain adaptation can be written as $p_\text{PolyRetro}(T|r)=\lambda p_\text{USPTO}(T|b-A-A-c) + (1-\lambda)\hat{p}(T)$ where $p_\text{USPTO}(T|b-A-A-c)$ is the pre-trained single-step model. Am I correct? The notations are somewhat confused because all density functions are denoted by $p$.
- Hyperparameter settings:
  - How to decide $\lambda$ without the validation set? I think one fold should be used for validation.
  - What $h$ is used for experiments? Is $h$ in the $\hat{p}(T)$ the same as used in Definition 4?

Concern (or Weakness) \#2: Methodology Novelty
- All sub-steps (except domain adaptation) are just filtering processes using a pre-trained single-step model. While the authors well-formulate the polymer retrosynthesis problem and well-explain the sub-steps, they seem somewhat straightforward. I am willing to give credit for this work because this is the first one (to my knowledge), but I am concerned about novelty in terms of methodology.

Concern (or Weakness) \#3: No qualitative study
- In retrosynthesis, top-k accuracy cannot reflect practical scenarios since multiple solutions can exist. Thus many papers often provide their success/failure cases and allow readers to evaluate them. In my opinion, such studies are critical in this field because it can provide additional insight into a chemist even though a prediction is wrong.

Concern (or Weakness) \#4: Baselines
- What is the support of $p(u|r)$? I'm curious about how RandomProposal works.
- I cannot be sure that RandomProposal and Transformer are good baselines. As I mentioned in the second concern, designing Poly Retro-USPTO is somewhat straightforward given the constraints of the proposed problem. Thus I think PolyRetro-USPTO is a more proper and stronger baseline.
- In the Appendix, there are stronger baselines than RandomProposal and Transformer. Why are they in the Appendix? I think they should be presented in the main paper.
- I want to know I correctly understand how mlp-retro and seq2seq-retro work. As far as I understand, they follow three sub-steps: (a) Given $r$, apply a pre-trained single-step retrosynthesis model (template-based mlp or template-free seq2seq) into $H-r-H$; (b) If the model obtains $H-B-b+c-C-H\rightarrow H-r-H$; (c) Then, the final monomers are $b-B-b$ and $c-C-c$ and the unit polymer is $b-B-C-c$. If I'm right, I wonder why mlp-retro is failed (because it is similar to PolyRetro-USPTO), and seq2seq-retro is not working for additional polymers as described in Appendix (because it just uses a pre-trained model). If I'm wrong, that can be a stronger baseline than RandomProposal/Transformer.
- Is Transformer trained using only one training fold from scratch?

---

Questions
- What is $\mathcal{R}_{m1,m2}$? I think its definition should be provided.
- Are the end-groups ($b$, $c$ in papers' definitions) of polymers often unknown? Or, are they not important? I wonder why only the repeat unit $r$ is given in the polymer retrosynthesis problem.
- Why the monomers should be symmetric? I think $b-B-b' + c'-C-c \rightarrow b-A-c + b'+c'$ can be another available candidate under the stability constraint (Definition 2).

---

Conclusion: While this paper focuses on an important problem, I have several concerns mentioned above, so  I think this paper is on the borderline.

---

> ### Author Response · Authors · 2020-11-21
> **Response to Reviewer 3**
>
> We thank the reviewer for the constructive comments! We have addressed the common issues on novelty and baselines in a separate comment. Please refer to "Response to common concerns on paper novelty and baselines" for more detail.
>
> Below we address the issues raised by the reviewer.
>
> **1. Domain adaptation.**
>
> In PolyRetro, to deal with the data scarcity issue in polymer retrosynthesis, we transfer the reaction/retrosynthesis knowledge from the small molecule domain to the polymer domain. Domain adaptation is needed here since the distribution of the types of reactions is not the same in these two domains. We agree with the reviewer that in general in small molecule cases, the reaction templates could be very dispersed, and using the empirical estimation from 52 reactions is not meaningful. However, in the case of condensation polymerizations, the space of types of reactions are more concentrated compared with small molecule reactions. Furthermore, the templates are less specific (extracted with radius=0) and we use a more general featurization of templates in our method. With more domain knowledge, one could replace the Gaussian kernel (Eqn. 8) with their own choice.
>
> Empirically, the results in the unit polymer recovery experiment show $\sim$5\% improvement of PolyRetro over PolyRetro-USPTO, which is not marginal. The marginal difference in the monomer recovery experiment is due to unseen templates for one-step retrosynthesis, which is discussed in the last two paragraphs in Section 6. If we have a perfect one-step retrosynthesis model, the gap will still be $\sim$5\% in the monomer recovery experiments.
>
> **2. Methodology Novelty.**
>
> The main novelties of this paper are:
> - From the material science perspective, we formulated an important problem and proposed the first learning-based framework solution.
> - From the machine learning perspective, we propose to use knowledge transfer and domain adaptation to approach a few-shot learning problem. Our learning-based framework for handling structural constraints are general and can be applied to other similar problems in machine learning as well (more detail in Appendix E).
>
> Please refer to the common comment for more discussions.
>
>
> **3. No qualitative study.**
>
> As a machine learning problem, our evaluation metric is the same as the one in small molecule retrosynthesis [1]. We understand the importance of qualitative study and we will plot success/failure examples and add discussions in the Appendix for reference in our revision.
>
> **4. Baselines.**
>
> For the Random Proposal baseline, the only difference between it and PolyRetro is that the unit polymer candidates are uniformly randomly sampled from the support of $p(u|r)$ instead of from (Eqn. 6). The support of $p(u|r)$ are the unit polymers satisfying the recursive constraint, which is computed in the polymer induction step in Section 5.1.
>
> We will replace the current baselines with the ones in Appendix D. For more discussions on those baselines, please refer to the common comment.
>
> mlp-retro and seq2seq-retro work in the following way:
> - Given repeat unit $r$, we complete it using the most common end-group pair in the whole dataset. We obtain $b-r-c$, in this particular case, $H-r-H$.
> - We apply one-step retrosynthesis model to obtain two reactants $B$ and $C$.
> - We check whether $B$ and $C$ satisfy the stability and synthesizability constraints. If yes, then $(B, C)$ is a monomer candidate.
>
> Both baselines do not check the recursive constraint, which is different from PolyRetro-USPTO or PolyRetro. PolyRetro-USPTO, on the other hand, shares most of the components with PolyRetro, and therefore should be considered as a variant of PolyRetro instead of a baseline.
>
> The transformer model is trained from scratch using one fold data.
>
> **5. Clarifications.**
>
> Hyperparameters.
> - $\lambda$ is decided by cross validation in training set.
> - $h=0.1$ (in Definition 4) which is obtained from running Retro* on ground-truth monomers.
> - $h=1$ (in Eqn. 8), we will use a different variable, thanks for pointing out!
>
> PolyRetro with domain adaptation.
> - When $u$ is uniquely determined by $T$, the reviewer is correct. Otherwise,
> $p_{PolyRetro}(u = \text{b-A-c}, T|r) = \lambda p_{USPTO}(\{u, u\}|\text{b-A-A-c}) + (1 - \lambda)\hat{p}(T)$
>
> Notations.
> - $\mathcal{R} _{m_1, m_2} = \mathcal{R} _{m_1} \cdot \mathcal{R} _{m_2}$
> - We will clarify the notation in the revision.
>
> Problem settings.
> - End-groups have a much less effect on the properties of polymers. Most of the time we care about the repeat units but not the end-groups. They are much less important.
> - Symmetric monomers are easier to form long chains, thus more "stable". Most of the known monomers for condensation polymers are symmetric.
>
>
> [1] Dai et al. "Retrosynthesis prediction with conditional graph logic network." Advances in Neural Information Processing Systems. 2019.

---

> > ### Comment · AnonReviewer3 · 2020-11-23
> > **Thank you for your detailed feedback**
> >
> > I appreciate your detailed feedback on my review. I agree with you about the **qualitative study,** **baselines,** **clarifications.** However, I am still concerned about domain adaptation and novelty.
> >
> > $$p_\text{PolyRetro}(u=b-A-c,T|r)=\lambda p_\text{uspto}(u,u|b-A-A-c) + (1-\lambda)\hat{p}(T)$$
> >
> > This optimal solution of the domain adaptation is the same as a simple probability modification based on template frequency $\hat{p}$ in the polymer dataset. Hence, I still think this adaptation technique is not novel from the machine learning perspective. Moreover, I'm not sure that the convex combination with the frequency-based template distribution effectively transfers knowledge into other few-shot benchmarks.
> >
> > Moreover, I'm not sure about the "learning-based" term. In PolyRetro, only $\hat{p}$ requires the polymer information, and other parts rely on pre-trained models ($p_\text{uspto}$ and Retro*) and filtering processes with problem constraints. In my opinion, frequency-based modeling $\hat{p}$ is not enough to claim the overall framework is a general learning-based solution.
> >
> > In summary, I fully agree with you about the first novelty, well-formulation for a new problem, but partially agree about the second and third one, the technical novelty, especially for domain adaptation. Hence, I want to keep my initial evaluation.

---

> > > ### Author Response · Authors · 2020-11-24
> > > **Further response to reviewer 3**
> > >
> > > Thanks for your kindly reply! We appreciate it and respect your thoughts. However, we still want to reiterate that,
> > > - Although the equations are simple, our proposed approach is still data-driven. The polymerization template distribution is learned from small molecule reactions, as well as the given polymer data.
> > > - We have achieved our goal - having a simple but effective solution. Increasing technical complexity is not our focus.
> > > - In case when there is more data available, the domain adaptation part of our framework can be replaced. For example, one can leverage deep generative approaches to model the distribution of the templates when there are more reaction supervisions in the target domain.

---

> ### Comment · Area_Chair1 · 2020-11-23
> **Feedback necessary**
>
> Dear Reviewer 3,
>
> Could you go over the response from the authors and see if it addresses your concern, and leave feedback? We will not be able to have interactive discussions after this Tuesday.
>
> Thanks, AC

---

### Official Review · AnonReviewer1 · 2020-10-27
**ICLR 2021 Conference Paper2253 AnonReviewer1**

**Rating:** 7
**Confidence:** 3

**Review:**

# summary #
This paper is concerning the retrosynthesis of polymerizations from monomer candidates.

A polymer is a very long chain of unit polymers. Therefore conventional retrosynthesis models do not suit well for polymerization since they focus on applications on small molecules.
The paper defines a problem of predicting a unit polymer and monomers, given a polymer repeat unit (Eq. 5).
A unit polymer can be decomposed into a polymer repeat unit and end-groups.
Thus the first component of the polyretro is a probabilistic proposal of the repeat-unit + end-groups using a one-step reaction model based on template reactions (Eq.6).
The second component of the polyretro is to choose combinations of monomers to synthesize the polymer unit (Eq.7).
Experimental validations reveal that the proposed polyretro algorithm can achieve 71% tope-30 recovery accuracy in unit polymer prediction, and 50% top-5 recovery accuracy for monomer prediction.

# comments #
I have been working on the retrosynthesis for smaller molecules and enjoys the paper reading.
I think this polymer challenge is quite new, and polymer synthesis is important for real applications.

Overall, the paper is well written and readability is satisfactory.
Problem definition of Eq. 5 seems straightforward but at the same time reasonable.
Technical sections (Sec 4. and 5) are easy to read: finds little difficulties to understand the main idea.

I cannot identify the technical novelty of the proposed polyretro method.
In Sec 3.2, "we use Retro*", but which parts of the algorithm are borrowed from Retro* and which parts are not?

Since this is a new challenge, it is difficult to assess the significance of the target recovery rates in experiments.
The Transformer baseline completely fails to recover targets in the experiments. This result itself is important to confirm that the typical retrosynthesis model is incapable.
However, such a ``weak baseline is not much informative to assess the significance of the proposed method.
The paper can be further empowered if the paper is equipped with additional experiments on a baseline method that actually works (not zero recovery scores). It can be a non-ML approach or a commercial software with block-box algorithms.

Minor issue:
In Fig. 1, the light pink of the polymer is invisible depending on the display/printer environments.

# evaluations #
(++) First attempt of retrosynthesis for polymer reactions, polymers are important in practical applications

(+) Readability is satisfactory in general.

(-) Technical novelty is unclear for me

(-) Significance of the target recovery scores is unclear to me

---

> ### Author Response · Authors · 2020-11-21
> **Response to Reviewer 1**
>
> We thank the reviewer for the constructive comments! We have addressed the common issues on novelty and baselines in a separate comment. Please refer to "Response to common concerns on paper novelty and baselines" for more detail.
>
> In addition, we want to clarify that the proposed PolyRetro method is general, and can be adapted to solve other similar problems in machine learning as well. Also, Retro* is only used in the Synthesizability check in Section 5.3. Please refer to Appendix E for more detail. We will move the discussion to the main text in the revised version.

---

> > ### Comment · AnonReviewer1 · 2020-11-23
> > **thank-you for your feedback**
> >
> > An additional experiments with stronger baseline is a pleasant surprise for me!! It is a nice follow-up experiment.
> > If the final camera-ready allows additional pages, it is nice to have this experiment in the main context.
> >
> > Also thank you for the clarification about Retrostar.
> >
> > I personally feel the feedback for the novelty does not bring significant new insights.
> >
> > Currently I keep my positive evaluation.

---

### Official Review · AnonReviewer2 · 2020-10-29
**A nice paper that tackles an important problem**

**Rating:** 6
**Confidence:** 3

**Review:**

This paper focuses on polymer retro synthesis problem. This is a novel problem and is very challenging because of the very small amounts of training data available (<100). They use reaction templates collected from small molecule reactions and formulate polymer retrosynthesis as a constrained optimization problem. The authors claim that this is the first learning method that takes constraints in polymer retrosynthesis problem.

The paper is well written and easy to follow. Overall I am positive about the paper as it proposes a new important tasks and provides a method that works well to tackle the task. The usage of reaction templates from small molecules as well as the ability to deal with with small amounts of training data is potentially useful and can serve as a model for few shot modeling in this domain for other tasks as well.

As for weaknesses,  I think the the baseline transformer performance of 0% top-50 recovery can be improved (and hence not a representation of true baseline). For example, can unsupervised pretraining be used in some way to improve its performance? Or can the small molecule reaction prediction data be used (aka in Molecule Transformer [1]) to seed the transformer model?

[1] Schwaller et al: Molecular Transformer: A Model for Uncertainty-Calibrated Chemical Reaction Prediction

---

> ### Author Response · Authors · 2020-11-21
> **Response to Reviewer 2**
>
> We thank the reviewer for the constructive comments! We will add another two baselines to the experiment section (currently in Appendix D) in the revised version. Please refer to the comment "Response to common concerns on paper novelty and baselines" for more detail.

---

> > ### Comment · AnonReviewer2 · 2020-11-24
> > **Thank you for the new baselines**
> >
> > Thank you for adding the new baselines. I think it makes the paper stronger.

---

### Official Review · AnonReviewer4 · 2020-10-29
**Recommendation to accept**

**Rating:** 6
**Confidence:** 3

**Review:**

##########################################################################

Summary:

This paper proposes a method for the retrosynthesis prediction of polymers. A challenge in this problem is the lack of synthetic data for polymers. The method attempts to leverage models for small molecule retrosynthesis predictions (where there is more abundant data), as well as domain specific constraints derived from the chemistry of a particular class of polymerization reactions. The method is shown to outperform some baselines that are commonly used in small molecule retrosynthesis.

##########################################################################

Reasons for score:

Overall, I vote for acceptance. I think the paper proposes a novel approach for polymer retrosynthesis that performs better than some of the baseline methods. However, I have some concerns, especially about the overall problem formulation of polymer retrosynthesis.

##########################################################################

Strengths:

*Paper is written clearly

*Interesting multistep method and applied constraints to convert a polymer repeat unit to monomers

*Some useful ablation studies

Weaknesses:

*Some concerns about the relevance of the overall problem formulation

*The method focuses on a particular type of polymerization reaction called condensation polymerization. Not sure how generalizable this method is to other common industrially relevant polymerization reactions, such as chain-growth type reactions

*The evaluation dataset is very small (52 examples)

##########################################################################

Questions and other comments:

*I wonder how practically relevant the overall polymer retrosynthesis problem is, at least in the way that it is currently presented. In small molecule synthesis (eg in medicinal chemistry or natural product synthesis) there is a well-defined target molecule containing the properties that we want, so it is useful to perform retrosynthesis on the target molecular structure to obtain the step by step synthesis procedure that describes how to create the target molecule from precursor starting materials. However, in polymer synthesis, there is a much less defined target polymer structure (we have a distribution of different polymer structures), and the synthetic procedure required to create the target polymer in condensation polymerization is typically a single step mixing of the monomer building blocks. In my opinion, the problem of converting the unit polymer to monomers and subsequently to precursor starting materials makes sense (and this is the aspect of the work that is similar to typical small molecule retrosynthesis because the unit polymer and monomers are small molecules). But the polymer induction part of the modeling, where we start with a given polymer repeat unit and convert it to the unit polymer, makes much less sense because in a real use case why wouldn’t you just perform the analysis (eg designing a new polymer or performing retrosynthetic analysis) directly on the unit polymer or monomer building blocks, with the knowledge that the target polymer structure is essentially just a repeated form of the unit polymer structure.

*How important is the polymer induction step to find the possible end groups? My intuition is that there are not that many possible end groups for condensation polymerization. It would be interesting to see the distribution of the average number of unit polymer candidates for each of the 52 examples.

*The additional seq2seq baseline for monomer proposal in Figure 7, where the repeat unit is converted to the unit polymer using a very simple heuristic (by adding the most common end group pairs) shows pretty competitive performance compared to the proposed PolyRetro model, and I think this simple heuristic is actually a very reasonable addition to the baselines.

*The PolyRetro model seems to have very marginal improvements over the PolyRetro-USPTO model, which seems to suggest a simple one step retrosynthesis model is already pretty good for the polymer induction step?

*Any thoughts about how to obtain more data for future developments in this area?

---

> ### Author Response · Authors · 2020-11-21
> **Response to Reviewer 4**
>
> We thank the reviewer for the constructive comments! We have addressed the common issues on novelty and baselines in a separate comment. Please refer to "Response to common concerns on paper novelty and baselines" for more detail.
>
> Here we address the issues raised by the reviewer.
>
> **1. Relevance of the problem formulation.**
>
> We agree with the reviewer that compared with the small molecule domain, in the polymer domain the target structure is usually less defined. However, since the properties of the polymer are largely decided by its chemical structures, and the repeat units are important factors in the forming of those chemical structures, there is a need to search for a polymer synthesis plan conditioned on the repeat unit: two polymers with repeat units similar in structure may lead to similar properties.
>
> Another reason for using repeat unit as a starting point because it is a conceptually simple representation of a polymer, while different unit polymers could lead to the same polymer (the same repeat unit with different feasible end-group pairs).
>
> **2. Generalizing to other polymerizations.**
>
> Our paper tackles condensation polymerization, which is one of the most common types of polymer reactions in the industry. The proposed method cannot be directly applied to other types of polymerization, as the constraints are not of the same type in different reactions. However, the framework can be adapted to handle other polymerizations as well, e.g., for the ionic polymerization, a type of chain-growth polymerization, the polymer induction in PolyRetro can be modified to model the ionic propagation step. We can generalize to the cases where we have an understanding of the recursion structure.
>
>
> **3. Polymer induction.**
>
> In theory, there could be an infinite number of possible end-groups. The polymer induction step is essential to propose end-group candidates that satisfy the constraints without the need to pre-define the end-groups. This method is more general and can generalize to unseen end-groups.
>
>
> **3. Data Scarcity.**
>
> In addition to the 52 examples, we collect an extra 100 polymers for expert evaluation. These 100 polymers come without ground-truth labels so we ask experienced chemists to score the predicted monomers from different algorithms. We find that our method is favored over other methods on most of the polymers. Please refer to Appendix D for more detail. We will move this part to the experiment section in our revision.
>
> In order to obtain more labeled data for future development, we are working on using NLP models to extract synthesis recipes of polymers from the chemistry literature. We hope that this field can attract more researchers as more data is available.
>
> **4. Experiments.**
>
> Please refer to the common comment on baselines for the discussion on seq2seq-retro.
>
> We want to clarify that the PolyRetro-USPTO baseline has the polymer induction step. The only difference with PolyRetro is that there is no domain adaptation in modeling $p(u|r)$ (Eqn. 6). However, the support of $p(u|r)$ is actually from the polymer induction step.
>
> The results in the unit polymer recovery experiment show $\sim$5\% improvement of PolyRetro over PolyRetro-USPTO, which is not marginal. The marginal difference in the monomer recovery experiment is due to unseen templates for one-step retrosynthesis, which is discussed in the last two paragraphs in Section 6.

---

### Author Response · Authors · 2020-11-21
**Response to common concerns on paper novelty and baselines**

We thank all the reviewers for their constructive feedback and helpful suggestions! Here we first address common concerns of the paper regarding the main contributions and the baselines in the experiments.

**1. Novelty of the paper (R1 and R3).**

Polymer retrosynthesis is an important problem related to small molecule retrosynthesis, a heated topic not only in drug design, but also in machine learning as well. Although these two problems looks similar, polymer retrosynthesis has its own challenges due to the "infinite" size of polymers and very limited number of available data points. To address these challenges, our paper for the first time

1) proposes a constrained optimization problem formulation for polymer retrosynthesis,

2) suggests a learning-based framework for solving the optimization problem, and

3) uses knowledge transfer from small molecule reactions and domain adaptation to tackle the data scarcity issue.

Furthermore, the proposed framework is general and can be applied to other similar problems in machine learning as well (refer to Appendix E for more details, we'll move the discussion to main text). In short, the main novelty of our paper lies in introducing/formalizing a new problem in science and proposing an general learning-based solution. We hope that our paper can inspire more work in the domain and benefit researchers in both communities.

**2. Baselines (All).**

Since the current Transformer baseline is directly trained on the training split of the 52 test polymers, the zero performance is expected as we know it is impossible to train a Transformer model on such a small number of data points. We were intended to use this baseline to make the point that knowledge transfer from small molecule reactions is needed for the few-shot learning setting. However, we do agree with the reviewers that this baseline itself is not enough for accessing the significance of the proposed method.

In the revised version we will include another two baselines, seq2seq-retro and mlp-retro, which directly predict the monomers from the repeat units without predicting the unit polymers. Both are based on existing one-step retrosynthesis methods. These two baselines are given additional advantages by knowing in advance the most common end-group pair among the 52 polymers, including the ones in the test split. Given the repeat unit $r$, the two baselines first obtain the "unit polymer" by completing the repeat unit using the most common end-group pair $a-r-b$, and then perform one-step retrosynthesis on the complete molecule $a-r-b$ to obtain the monomer candidates. The model and training details of these two baselines are described below.

- **seq2seq-retro**: seq2seq one-step retrosynthesis model trained on USPTO dataset, and then fine-tuned on the training split of 52 polymers.
- **mlp-retro**: mlp templated-based retrosynthesis model trained on USPTO dataset, cannot be fine-tuned since it is template-based.

As shown in Figure 7 in Appendix D, both baselines achieve non-zero performance in the monomer recovery task, and seq2seq-retro even achieves very competitive performance compared with PolyRetro. For seq2seq-retro, despite having very good performance, it is also worth noting that

1) The handling of the stability and synthesizability constraint is the same with PolyRetro. The only difference is in the handling of the recursive constraint, where seq2seq-retro has an unfair advantage.

2) In the extra human-evaluation experiment of 100 additional polymers described in Appendix D, the seq2seq-retro model is not performing well.

For all polymers whose repeat units are much larger than the ones in the training split, seq2seq-retro predicts the same monomer. This is probably due to seq2seq models do not generalize well to longer and unseen sequences. At the same time, PolyRetro, which is template-based, does not suffer from such issue.

We will move this part (both additional baselines and human-evaluation experiments) to our experiment section in the main text and discuss them in detail since now we have one extra page for writing.

---

### Comment · Area_Chair1 · 2020-11-23
**The end of the interactive discussion phase approaching**

Dear Reviewers,

The authors have provided detailed responses to your comments. Could you please go over the responses from the reviewers and provide feedback since the authors can have interactions with you only by this Tuesday (24th)?. I sincerely thank you for your service in reviewing for ICLR.

Thanks,
Area Chair

---

### Decision · Program_Chairs · 2021-01-07
**Final Decision**

**Decision:**

Reject

**Comment:**

This paper proposes a novel problem of polymer retrosynthesis, and a method to solve it. The authors formally define the polymer retrosynthesis optimization problem as a constrained problem to identify the monomers and the unit polymer, with the recursive and stability constraints. Further, since the main challenge with polymer retrosynthesis is the extremely scarce training data, the authors propose a domain adaptation technique that can utilize a single-step retrosynthesis model trained on a large amount of data. The authors also use Retro* [Chen et al. 20] for synthesizability check of the monomers. The proposed method, PolyRetro, is validated against few naive baselines for top-k recovery performance, and is shown to outperform them.

All reviewers found the problem of polymer retrosynthesis tackled to be important as well as novel, and the paper to be very well-written. However, all reviewers had a common concern on the limited technical novelty and meaningless baselines that makes it difficult to evaluate the significance of the results. Some of the reviewers were also concerned with the insignificant performance gain with the proposed domain adaptation technique (PolyRetro vs. PolyRetro-USPTO in Figure 4), and its limited applicability to a condensation polymerization. The authors provided new results with more baselines, which fine-tune the single-step retrosynthesis model (MLP, seq-to-seq) trained on USPTO.

The below is the summary of pros and cons:

**Pros**
- The tackled polymer retrosynthesis problem is novel and practically important.
- The proposed problem formulation and constraints are interesting and make sense.
- The paper is well-written and easy to follow even for non-domain experts.

**Cons**
- The proposed solution with recursive and stability constraints is rather straightforward, as well as the use of Retro* for screening out the monomers.
- The domain adaptation technique, which is advertised as an important contribution to combat extreme data scarcity, is both straightforward and yields small performance gain.
- The baselines in the original version of the paper are simply meaningless strawmans, and the new baseline (seq2seq-retro) in Section D of the Appendix seems quite strong, making it difficult to validate the effectiveness of the proposed method.

The paper received split reviews, with three leaning toward acceptance and one leaning toward rejection. After the interactive discussion period with the authors, the reviewers had an in-depth discussion, where all reviewers agreed that the technical novelty or contribution to the general machine learning field, or general applicability to polymer synthesis is limited. The reviewers did not reach a consensus, which makes the paper a borderline case, and after the discussion with the program chairs, we decided to reject the paper due to the unresolved concerns.

I believe that the proposed problem-specific solution is adequate, although it has little technical novelties, since this is an application paper. However what is more problematic is the inconclusive experimental validation results due to lack of meaningful baselines. I suggest the authors to compare against seq2seq retro + Retro* in order to properly validate the effectiveness of the proposed method. Also, results in Figure 7, or the polymerization examples in Section A of the appendix should be incorporated into the main paper. I also suggest that the authors drop domain adaptation from the title since it constitutes a small part of the method and thus is misleading.